# How has the emergence of the Omicron SARS-CoV-2 variant of concern influenced worry, perceived risk and behaviour in the UK? A series of cross-sectional surveys

Louise E Smith ![ORCID] [1,2] Henry WW Potts,[3] Richard Amlôt,[2,4] Nicola T Fear ![ORCID] [1,5] Susan Michie,[6] G James Rubin[1,2]

For numbered affiliations see end of article.

**Correspondence to**
Dr Louise E Smith;
louise.e.smith@kcl.ac.uk

## ABSTRACT

**Objectives** To investigate changes in beliefs and behaviours following news of the Omicron variant and changes to guidance understanding of Omicron-related guidance, and factors associated with engaging with protective behaviours.

**Design** Series of cross-sectional surveys (1 November to 16 December 2021, five waves of data collection).

**Setting** Online.

**Participants** People living in England, aged 16 years or over (n=1622–1902 per wave).

**Primary and secondary outcome measures** Levels of worry and perceived risk, and engagement with key behaviours (out-of-home activities, risky social mixing, wearing a face covering and testing uptake).

**Results** Degree of worry and perceived risk of COVID-19 (to oneself and people in the UK) fluctuated over time, increasing slightly around the time of the announcement about Omicron (p<0.001). Understanding of rules in England was varied, ranging between 10.3% and 91.9%, with people overestimating the stringency of the new rules. Rates of wearing a face covering and testing increased over time (p<0.001). Meeting up with people from another household decreased around the time of the announcement of Omicron (29 November to 1 December), but then returned to previous levels (p=0.002). Associations with protective behaviours were investigated using regression analyses. There was no evidence for significant associations between out-of-home activity and worry or perceived risk (COVID-19 generally or Omicron-specific, p≥0.004; Bonferroni adjustment p<0.002 applied). Engaging in highest risk social mixing and always wearing a face covering were associated with worry and perceived risk about COVID-19 (p≤0.001). Always wearing a face covering in shops was associated with having heard more about Omicron (p<0.001).

**Conclusions** Almost 2 years into the COVID-19 outbreak, the emergence of a novel variant of concern only slightly influenced worry and perceived risk. The main protective behaviour (wearing a face covering) promoted by new guidance showed significant re-uptake, but other protective behaviours showed little or no change.

## STRENGTHS AND LIMITATIONS OF THIS STUDY

⇒ Rapid data collection, reporting on beliefs and behaviours immediately following news of the emergence of the Omicron variant of concern.

⇒ Large sample size, and repeated questions, allow for precise prevalence estimates and investigation of longer-term trends.

⇒ Data are self-reported and may, therefore, represent an overestimation of engagement with protective behaviours.

⇒ Data are cross-sectional, and we cannot imply the direction of associations.

⇒ We are unsure of the representativeness of the beliefs and behaviours of people who sign up to take part in online surveys.

## INTRODUCTION

The Omicron variant of SARS-CoV-2 was reported to the WHO on 24 November 2021 and was designated by the WHO as a variant of concern on 26 November 2021.[1] Since this date, it has attracted substantial media coverage.[2 3] The emergence of the Omicron variant presented policymakers, and society more generally, with a dilemma. What action should be taken in the face of a rapidly spreading infection, the severity of which is unclear? The UK witnessed intense debate around this question, with disagreements being played out across the national press, in the House of Commons and in academic articles. In the early stages of the COVID-19 pandemic, the emergence of the original SARS-CoV-2 virus prompted similar controversy and led to modest increases in levels of worry among the UK public, with 40% engaging in recommended respiratory and hand hygiene behaviours, and 14% reducing the number of people that they met, a behaviour that had not then been officially recommended.[4]

England removed legal COVID-19 mandates to wear a face covering and physically distance on 19 July 2021.[5] This was followed by decreases in rates of protective behaviour.[6] In response to the Omicron variant, the UK Prime Minister, English Chief Medical Officer and Government Chief Scientific Advisor held a press conference on 27 November 2021, the same day the first UK cases were reported,[7] in which new measures were announced.[8] These were implemented from 30 November.[9] They included making face coverings compulsory in shops and on public transport, and requiring all international arrivals to take a PCR test within 2 days of arriving in the UK and self-isolating until they received a negative test result.[5 6] Recommendations for all members of the public to use lateral flow tests regularly, and before meeting other people (epitomised by the slogan 'lateral flow before you go' used in the Devolved Administrations[10]) were retained and reiterated.

As more evidence about the rapid spread of the Omicron variant appeared, on 8 December 2021, further measures were announced as part of the UK's 'Plan B', with face coverings becoming compulsory in most public indoor venues (apart from hospitality), vaccine passports becoming mandatory in specific settings and people being asked to work from home where possible.[11] These changes came into effect on 13 December 2021. On 27 December, the government announced no new restrictions for England before the end of the year.[12]

Throughout the pandemic, concern has been raised that public adherence to rules may wane over time.[13] Nonetheless, changes in rules have consistently caused changes in behaviour.[14] Research conducted during the COVID-19 and the 2009 H1N1 pandemics indicated that engagement with protective behaviours was associated with having heard more about the pandemic,[4 15] and increased worry about, and perceived risk of, infection.[16 17] Public fears are known to be greater when risks are novel and uncertain.[18] While the risks of COVID-19 were familiar to the public, the new variant represented a possible new source of public worry that may have affected behaviour.

In this study, we investigated whether beliefs about COVID-19 and engagement with protective behaviours changed in the first 3 weeks of the emergence of the Omicron variant. We measured understanding of new guidance and satisfaction with the government response to Omicron. We also investigated whether engaging with protective behaviours was associated with amount heard about Omicron, worry (about COVID-19 generally and Omicron specifically) and perceived risk (of COVID-19 generally and Omicron specifically).

## METHODS
### Design
A series of online cross-sectional surveys have been conducted by BMG Research then Savanta (both Market Research Society company partners) since January 2020 on behalf of the English Department of Health and Social Care, and analysed by the COVID-19 Rapid Survey of Adherence to Interventions and Responses (CORSAIR) research team.[19] For these analyses, we used data collected in five waves: wave 61 (1–4 November 2021), wave 62 (15–17 November), wave 63 (29 November–2 December), an ad hoc wave added to the series to assess responses to Omicron (6–8 December; wave 63.5) and wave 64 (13–16 December).

Questions in each wave asked about behaviour over the previous week. Data collection for wave 63 took place after the first news about Omicron and the announcement of new rules. It spanned a longer period before (8 days), and a shorter period after (3 days), the rules came into force (30 November 2021; see online supplemental material figure 1 for a timeline). The added survey (wave 63.5) was issued after the emergence of Omicron, but encompassed a shorter period before (1 day), and a longer period after (9 days), the new rules came into force. Wave 64 data collection started on the same day as further rules ('plan B') came into force (13 December 2021; rules announced on 8 December). Wave 64 data therefore encompasses a longer period before (7 days), and after shorter period after (4 days), plan B rules came into force.

### Participants
Participants were recruited from a pool of people who had signed up to take part in online surveys (known as online research panels). Participants were eligible to take part if they were aged 16 years or over and lived in the UK. Non-probability sampling (quotas based on age and sex (combined), and region) was used to ensure the sample was broadly similar to the UK general population. After completing the survey, participants were unable to take part in the subsequent three waves of data collection. Participants were reimbursed in points which could be redeemed in cash, gift vouchers or charitable donations (up to 70p per survey).

We report figures for England only as the four nations of the UK implemented different changes for Omicron. We excluded participants in Wave 63.5 who completed the survey after the 8 December Government press conference began (n=58).

### Study materials
Unless otherwise specified, participants answered all items.

#### Worry and perceived risk
Participants were asked 'overall, how worried are you about coronavirus' on a five-point scale from 'not at all worried' to 'extremely worried'. They were also asked 'to what extent you think coronavirus poses a risk to…' them personally and people in the UK, on a five-point scale from 'no risk at all' to 'major risk'. From wave 63.5, participants were also asked congruent questions about their worry about, and perceived risk of, Omicron. The items asked participants 'Thinking about the Omicron

variant, how worried are you about this specific variant of coronavirus?' and 'to what extent you think this specific variant of coronavirus poses a risk...'.

Worry and perceived risk (to oneself, others in the UK) were coded into separate binary variables (worry: very and extremely worried, vs somewhat, not very, and not at all worried; perceived risk: major and significant risk, vs moderate, minor and no risk at all).

### Behaviours

Participants were asked how many times in the last week they had done each of a list of 20 activities including shopping for groceries/pharmacy, shopping for other items, providing help or care for a vulnerable person, meeting up with friends or family that they did not live with, going to a restaurant, café or pub, using public transport or a taxi/minicab and going out to work (number of days). Responses were capped at 30; going out to work was capped at 7.

Participants who indicated that they had met up with friends or family from another household were asked a series of follow-up questions about the setting and number of people involved in their most recent meeting in the past 7 days. We derived a measure categorising the risk of transmission involved in a participant's most recent instance of social mixing.[14] We were unable to calculate this measure for five participants due to missing data.

Participants who indicated that they had visited a shop, hospitality venue or used public transport or a minicab were asked whether they wore a face covering while doing so. Response options were 'yes—on all occasions', 'yes—on some occasions' and 'no, not at all'. We categorised people as wearing a face covering all the time, versus sometimes or not at all.

We asked participants when they last took a test for coronavirus. We categorised people as having tested if they indicated that they took their most recent test in the last week.

### Amount heard about omicron

From wave 63.5, participants were asked to indicate 'how much, if anything, have you seen or heard about the new Omicron variant of coronavirus that was first detected in southern Africa?' on a four-point scale from 'I have not seen or heard anything' to 'I have seen or heard a lot'.

### Satisfaction with government response

Participants in wave 63.5 onwards were asked to what extent they agreed or disagreed that 'The Government was putting the right measures in place to protect the UK public from the Omicron variant of coronavirus', you 'have enough information from the Government and other public authorities on the symptoms associated with the Omicron variant of coronavirus', and you 'have enough information from the Government and other public authorities on how effective current vaccines are against the Omicron variant of coronavirus' on a five-point scale from strongly agree to strongly disagree.

### Understanding of new rules

From wave 63.5, participants living in England were asked to indicate whether a series of nine statements about rules brought in to prevent the spread of Omicron were true, false or they did not know. A tenth statement was added for wave 64. Statements included items about wearing a face covering in different locations (in shops, on public transport, in hospitality venues), self-isolation and out-of-home behaviour.

### Sociodemographic factors

We measured participants' age in years, sex, employment status, socioeconomic grade, highest educational or professional qualification, ethnicity, their first language, COVID-19 vaccination status, whether there was a dependent child in the household, whether they were at high risk for COVID-19,[20] whether a household member had a chronic illness, and whether they thought they had previously, or currently, had COVID-19 (recoded to a binary variable: 'I've definitely had it, and had it confirmed by a test' and 'I think I've probably had it', vs 'I don't know whether I've had it or not', 'I think I've probably not had it' and 'I've definitely not had it'). Participants were also asked to report their full postcode, from which geographical region and indices of multiple deprivation were determined.[21]

To measure financial hardship, participants were asked to what extent in the past 7 days they had been struggling to make ends meet, skipping meals they would usually have and were finding their current living situation difficult (Cronbach's α=0.84).

### Patient and public involvement

Lay members served on the advisory group for the project that developed our prototype survey material; this included three rounds of qualitative testing.[22] Due to the rapid nature of this research, the public was not involved in the further development of the materials during the COVID-19 pandemic.

### Power

A sample size of 1600 per wave allows a 95% CI of approximately plus or minus 2% for the prevalence estimate for a survey item with an overall prevalence of 50%.

### Analysis

Unless otherwise specified, answers of 'don't know' were coded as missing.

We limited analyses investigating non-essential workplace attendance to participants who reported being in full employment, part employment or self-employment, and who indicated that they could work from home full time. Questions about wearing a face covering were only asked to people who reported having completed that activity in the past 7 days. Therefore, analyses were restricted to those who reported having been in shops, on public transport, and in hospitality venues in the last week.

We plotted worry and perceived risk, and behaviours by survey wave. For uptake of testing, we plotted two lines,

**Table 1** Respondent characteristics

| Attribute | Level | Wave 61, % (n) (total n=1833) | Wave 62, % (n) (total n=1902) | Wave 63, % (n) (total n=1743) | Wave 63.5, % (n) (total n=1622) | Wave 64, % (n) (total n=1841) | P value |
|---|---|---|---|---|---|---|---|
| Sex | Male | 46.8 (853) | 47.2 (893) | 46.7 (812) | 45.8 (741) | 47.0 (862) | 0.94 |
| | Female | 53.2 (968) | 52.8 (999) | 53.3 (925) | 54.2 (878) | 53.0 (973) | |
| Age | Range 16 to >90 years | M=48.7, SD=19.2 | M=47.8, SD=18.8 | M=49.1, SD=18.2 | M=47.7, SD=18.4 | M=47.7, SD=18.8 | 0.07 |
| Employment status | Not working | 46.8 (844) | 44.8 (840) | 45.5 (786) | 44.0 (707) | 44.7 (813) | 0.54 |
| | Working | 53.2 (959) | 55.2 (1033) | 54.5 (943) | 56.0 (899) | 55.3 (1005) | |
| Index of multiple deprivation | First (least) to fourth quartile (most deprived) | M=2.7, SD=1.0 | M=2.7, SD=1.0 | M=2.7, SD=1.0 | M=2.8, SD=1.0 | M=2.7, SD=1.0 | 0.62 |
| Highest educational or professional qualification | Less than degree | 65.4 (1198) | 67.1 (1277) | 66.8 (1165) | 65.9 (1069) | 67.5 (1243) | 0.63 |
| | Degree or higher | 34.6 (635) | 32.9 (625) | 33.2 (578) | 34.1 (553) | 32.5 (598) | |
| Ethnicity | White British | 82.2 (1498) | 82.7 (1563) | 84.2 (1460) | 82.4 (1329) | 82.0 (1505) | 0.09 |
| | White other | 6.1 (111) | 5.1 (96) | 5.5 (96) | 5.1 (82) | 4.5 (83) | |
| | Black and minority ethnicity | 11.7 (214) | 12.2 (231) | 10.2 (177) | 12.5 (202) | 13.5 (247) | |
| Vaccination status | Not vaccinated | 10.7 (195) | 14.4 (269) | 13.1 (226) | 13.4 (215) | 13.6 (248) | 0.03 |
| | 1 dose | 5.4 (99) | 6.0 (112) | 5.2 (89) | 6.5 (104) | 6.3 (115) | |
| | 2 doses or more | 83.9 (1528) | 79.7 (1493) | 81.7 (1411) | 80.2 (1291) | 80.1 (1463) | |

Where percentages do not sum to 100%, this is due to rounding errors.

including and excluding those whose most recent test was a PCR test and who did not know their most recent test type. To investigate change over time, we used $\chi^2$ analyses (categorical data), one-way analysis of variances (continuous data) and Kruskal-Wallis tests (skewed continuous data).

We present descriptive statistics of participants' understanding of the new rules brought in in response to Omicron and satisfaction with the government response.

To investigate associations with engagement with protective behaviours, we used data collected 6 to 8 December 2021 (wave 63.5) and 13 to 16 December 2021 (wave 64) separately as we hypothesised that people's views and behaviour were likely to change due to the fast-moving nature of the spread of Omicron. We used negative binomial regression analyses (to account for skewed outcomes) to investigate associations with out-of-home activities (going out shopping, going to the workplace). For these analyses, we summed the number of times participants reported going out shopping for groceries/pharmacy and other items, to give a total number of times gone shopping. We ran one model including only sociodemographic factors; a second that additionally included amount heard about Omicron, and either perceived worry about COVID-19, perceived risk of COVID-19 to oneself, or perceived risk of COVID-19 to people in the UK; and a third that additionally included Omicron-specific worry

or perceived risk. For these analyses, we report adjusted incidence rate ratios (aIRRs).

For binary outcomes (risky social mixing: highest risk social mixing, vs other; always wearing a face covering in shops: wearing a face covering on all occasions, vs other; wearing a face covering in hospitality venues: wearing a face covering on all occasions, vs other), we used logistic regression analyses. Sociodemographic factors were entered as block one. Amount heard about Omicron and either worry about COVID-19, perceived risk of COVID-19 to oneself, or perceived risk of COVID-19 to people in the UK were entered as block two. Omicron-specific worry, perceived risk to self or perceived risk to people in the UK were entered as block three. For these analyses, we report adjusted odds ratios (aORs).

To account for the large number of analyses, we used a Bonferroni correction. For analyses investigating changes in beliefs and behaviour over time, we set significance at p<0.003 (n=22). For regression analyses, we set significance at p<0.002 (n=28).

## RESULTS
### Respondent characteristics
A total of 8941 responses were included in analyses (wave 61, n=1833; wave 62, n=1902; wave 63, n=1743; wave 63.5,

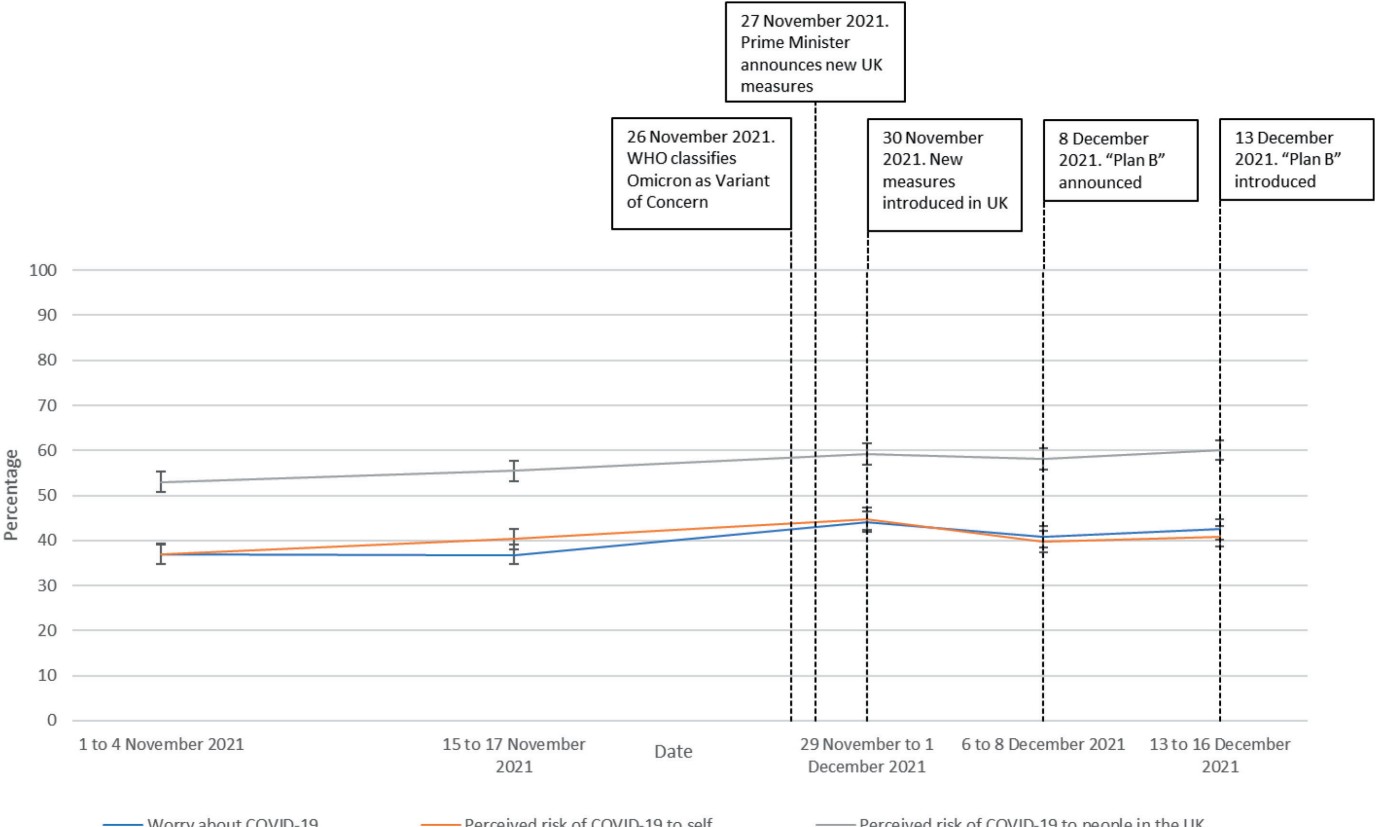

**Figure 1** Perceived worry about, and risk of, COVID-19 between 1 November 2021 and 16 December 2021.

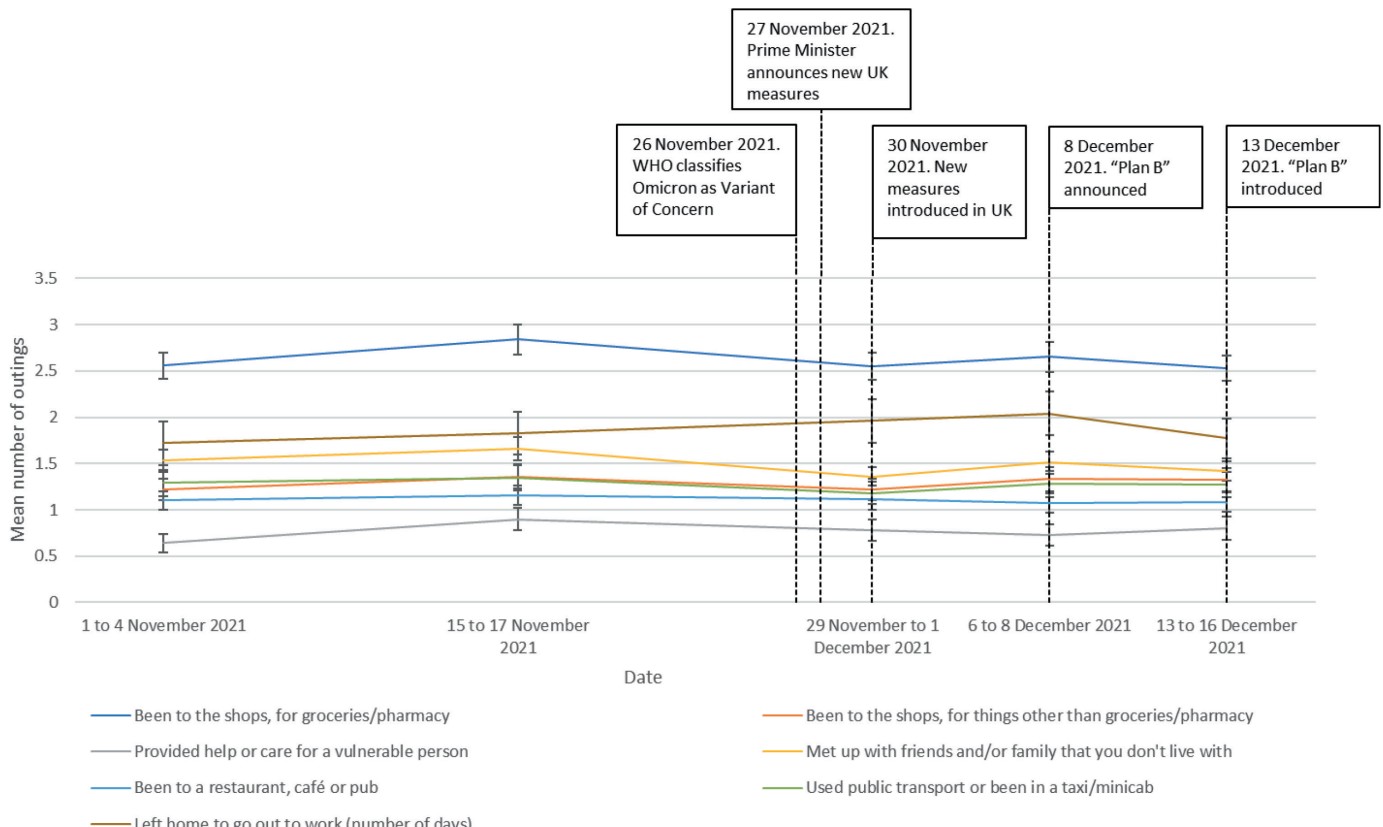

**Figure 2** Out-of-home activity, between 1 November 2021 and 16 December 2021.

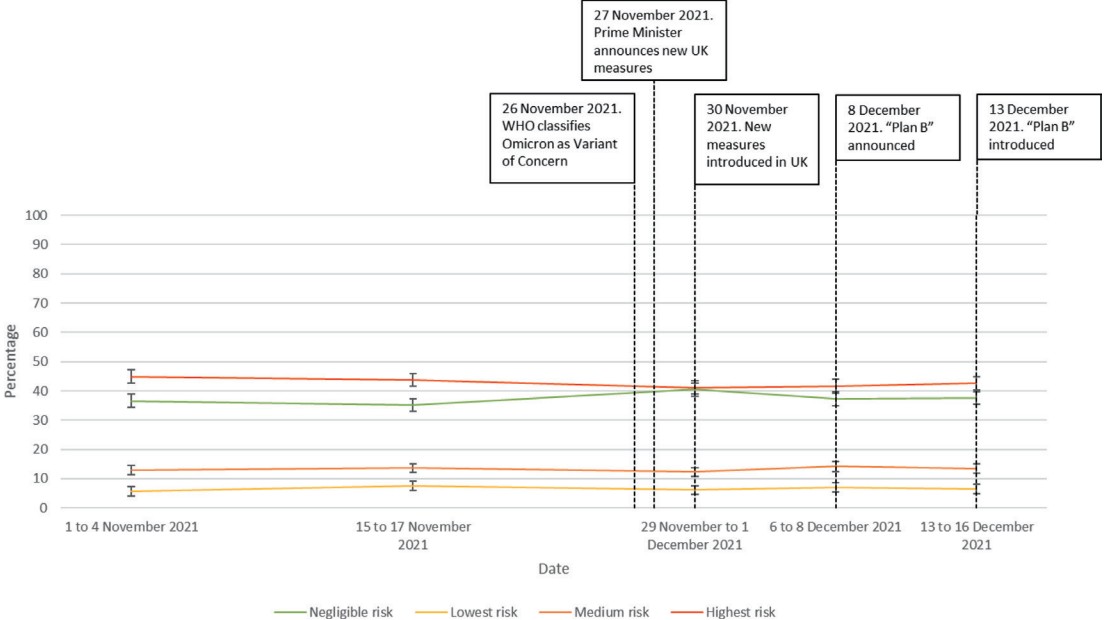

**Figure 3** Risky social mixing, between 1 November 2021 and 16 December 2021.

n=1622; wave 64, n=1841). Respondents were slightly more likely to be women, white, and educated to degree level or higher compared with the general population (table 1).[23][24] There was a significant difference in uptake of vaccination ($\chi^2(8)=17.0$, p=0.03). In practice, there were small differences between waves, with percentages differing at most by 4.2%.

### Beliefs and behaviours over time

Perceived worry about, and risk of, COVID-19 fluctuated over time, with worry, perceived risk to self and perceived risk to people increasing slightly around the time of the announcement about the Omicron variant, then returning to pre-Omicron levels (worry ($F(4,8921)=10.08$, p<0.001); perceived risk to self ($F(4,8857)=7.10$, p<0.001);

perceived risk to people in UK ($F(4,8854)=5.12$, p<0.001); figure 1).

Between 1 November and 16 December 2021, reported rates of meeting up with people from another household changed ($H(4) =17.4$, n=8941, p=0.002; figure 2). This change was driven by a decrease in reported rates in data collected on 29 November to 1 December 2021 (wave 63, around the time of the announcement of Omicron) compared with the previous survey wave. Providing help or care for a vulnerable person also changed between 1 November and 16 December 2021 ($H(4)=17.0$, n=8941, p=0.002), with this change being driven by an increase in reported rates in data collected on 15 to 17 November 2021 compared with the previous survey wave. There

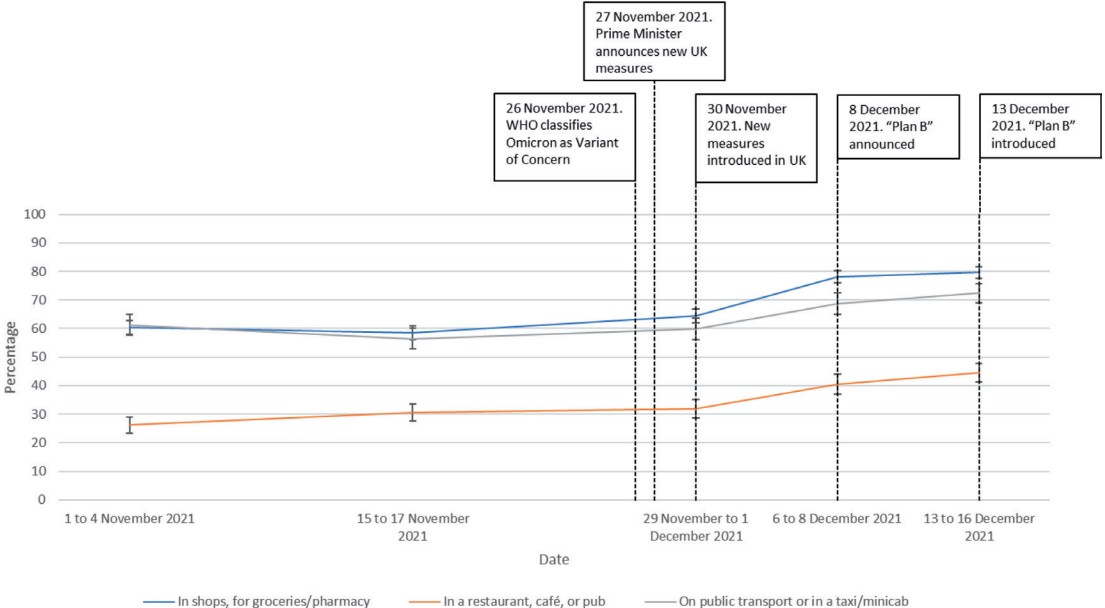

**Figure 4** Always wearing a face covering, between 1 November 2021 and 16 December 2021.

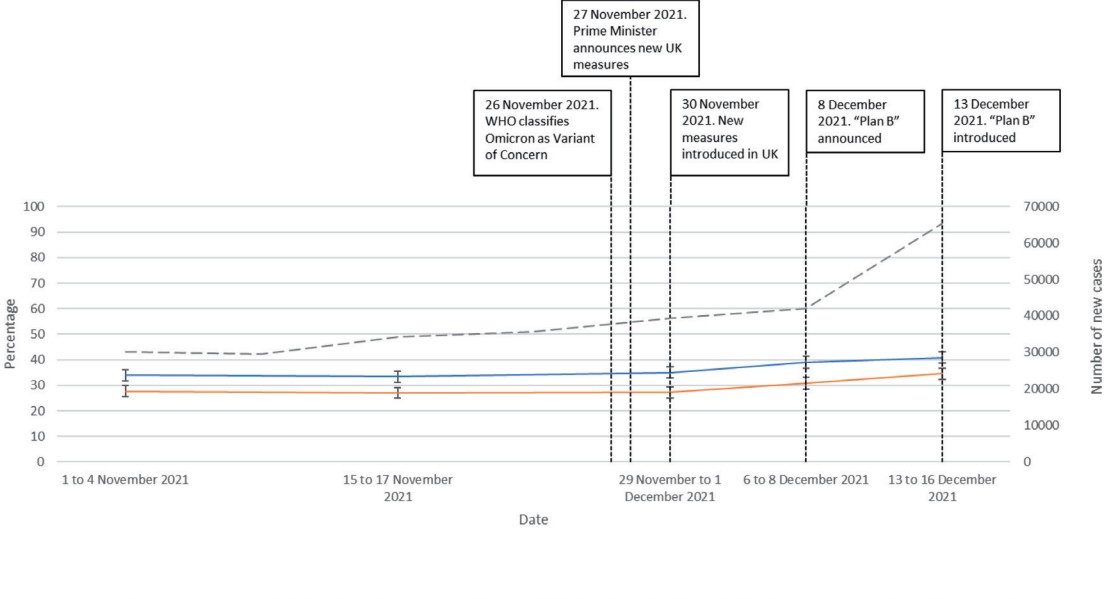

**Figure 5** Uptake of testing, between 1 November 2021 and 16 December 2021. The dashed line shows the 7-day average for new cases in England.

were no other significant changes in out-of-home activity over time (been to the shops, for groceries/pharmacy ($H(4)=7.5$, n=8941, p=0.11); been to the shops, for things other than groceries/pharmacy ($H(4)=8.4$, n=8941, p=0.08); been to a restaurant, café or pub ($H(4)=7.0$, n=8941, p=0.14); used public transport or been in a taxi/minicab ($H(4)=1.1$, n=8941, p=0.90); left home to go to out to work (number of days) ($H(4)=4.3$, n=1904, p=0.36).

There were no differences in social mixing over time, stratified by risk of transmission ($H(4)=8.9$, p=0.06; figure 3).

Rates of always wearing a face covering increased over time in all settings (in shops for groceries/pharmacy ($\chi^2(4)=286.0$, n=7815, p<0.001); in a restaurant, café or pub ($\chi^2(4)=90.9$, n=4497, p<0.001); on public transport or in a taxi/minicab ($\chi^2(4)=50.8$, n=3310, p<0.001); figure 4).

Rates of testing increased over time (whole sample, $\chi^2=33.2$ (4), n=8780, p<0.001; excluding people whose most recent test was a PCR test or who did not know what their most recent test type was, $\chi^2=32.4$ (4), n=7912, p<0.001; figure 5).

### Omicron worry, perceived risk and amount heard

A total of 39.0%–42.7% of people reported being very or extremely worried about the Omicron variant (table 2, figure 1). More people (44.9%–46.4%) perceived a major or significant risk of Omicron to themselves, with 56.7%–61.4% of respondents perceiving a major or significant risk of Omicron to people in the UK. When applying a Bonferroni correction, there was no significant difference in Omicron worry or risk between Wave 63.5 and wave 64 data (worry: $F(1,3417)=4.74$, p=0.03; perceived risk to self: $F(1,3371)=0.75$, p=0.39; perceived risk to people in the UK: $F(1,3391)=7.67$, p=0.006).

### Understanding of new rules

Understanding of the new rules introduced in response to Omicron was varied (table 3). Understanding of rules requiring behaviour was good (around 80%+ correct, 90%+ correct on some rules). However, other items were answered incorrectly by most people, in the direction of believing that the rules were stricter than was the case. For some items (wearing a face covering in hospitality venues and all crowded and enclosed spaces), the percentage overestimating the rules increased from wave 63.5 to wave 64. From 13 December 2021, people were asked to work from home if possible. This was the only rule that changed between survey waves, with high recognition in the latter wave.

Fewer than half of respondents agreed that the government were putting the right measures in place to protect the UK public from Omicron, with around half agreeing that they had enough information about the symptoms of the Omicron variant and the effectiveness of vaccines against Omicron variant (table 4). Most people agreed that they had enough information about what to do to prevent the spread of Omicron.

### Factors associated with engaging with protective behaviours

There were no significant associations between out-of-home activity and amount heard about Omicron, perceived worry (COVID-19 generally or Omicron specifically) or perceived risk (to oneself or people in UK, COVID-19 generally or Omicron specifically; table 5). There were no associations with sociodemographic characteristics, with the exception of greater financial hardship being associated with going out shopping for items other than groceries/pharmacy (see online supplemental table 1).

**Table 2** Perceived worry about, and risk of, Omicron variant

| | Thinking about the Omicron variant, how worried are you about this specific variant of coronavirus? | | | | | Still thinking about the Omicron variant, to what extent do you think this specific variant of coronavirus poses a risk to you personally | | | | | Still thinking about the Omicron variant, to what extent do you think this specific variant of coronavirus poses a risk to people in the UK | | | | | How much, if anything, have you seen or heard about the new Omicron variant of coronavirus that was first detected in southern Africa? | | | | |
|---|---|---|---|---|---|---|---|---|---|---|---|---|---|---|---|---|---|---|---|---|
| | Wave 63.5 (total n=1622) | | Wave 64 (total n=1841) | | | | Wave 63.5 (total n=1622) | | Wave 64 (total n=1841) | | | | Wave 63.5 (total n=1622) | | Wave 64 (total n=1841) | | | | Wave 63.5 (total n=1622) | | Wave 64 (total n=1841) | |
| | n | % (95% CI) | n | % (95% CI) | | | n | % (95% CI) | n | % (95% CI) | | | n | % (95% CI) | n | % (95% CI) | | | n | % (95% CI) | n | % (95% CI) |
| Extremely worried | 229 | 14.1 (12.5 to 15.9) | 316 | 17.2 (15.4 to 18.9) | Major risk | 334 | 20.6 (18.7 to 22.6) | 427 | 23.2 (21.3 to 25.1) | 414 | 25.5 (23.4 to 27.7) | 533 | 29.0 (26.9 to 31.0) | I have seen or heard a lot | 512 | 31.6 (29.3 to 33.9) | 683 | 37.1 (34.9 to 39.3) |
| Very worried | 395 | 24.4 (22.3 to 26.5) | 461 | 25.0 (23.1 to 27.0) | Significant risk | 369 | 22.7 (20.7 to 24.8) | 413 | 22.4 (20.5 to 24.3) | 478 | 29.5 (27.3 to 31.8) | 583 | 31.7 (29.5 to 33.8) | I have seen or heard a fair amount | 774 | 47.7 (45.3 to 50.2) | 814 | 44.2 (41.9 to 46.5) |
| Somewhat worried | 599 | 36.9 (34.6 to 39.3) | 628 | 34.1 (31.9 to 36.3) | Moderate risk | 452 | 27.9 (25.7 to 30.1) | 489 | 26.6 (24.5 to 28.6) | 441 | 27.2 (25.0 to 29.4) | 438 | 23.8 (21.8 to 25.7) | I have seen or heard a little | 303 | 18.7 (16.8 to 20.7) | 300 | 16.3 (14.6 to 18.0) |
| Not very worried | 267 | 16.5 (14.7 to 18.4) | 265 | 14.4 (12.8 to 16.0) | Minor risk | 329 | 20.3 (18.4 to 22.3) | 386 | 21.0 (19.1 to 22.8) | 198 | 12.2 (10.7 to 13.9) | 207 | 11.2 (9.8 to 12.7) | I have not seen or heard anything | 27 | 1.7 (1.1 to 2.4) | 36 | 2.0 (1.3 to 2.6) |
| Not at all worried | 109 | 6.7 (5.6 to 8.1) | 150 | 8.1 (6.9 to 9.4) | No risk at all | 80 | 4.9 (3.9 to 6.1) | 94 | 5.1 (4.1 to 6.1) | 43 | 2.7 (1.9 to 3.6) | 58 | 3.2 (2.4 to 3.9) | | | | | |
| Don't know | 23 | 1.4 (0.9 to 2.1) | 21 | 1.1 (0.7 to 1.6) | Don't know | 58 | 3.6 (2.7 to 4.6) | 32 | 1.7 (1.1 to 2.3) | 48 | 3.0 (2.2 to 3.9) | 22 | 1.2 (0.7 to 1.7) | Don't know | 6 | 0.4 (0.1 to 0.8) | 8 | 0.4 (0.1 to 0.7) |

**Table 3** Endorsement of rules introduced in response to Omicron

| The government has issued new rules on how people should act to help prevent the spread of the omicron variant of coronavirus. Please tell us, for the following options, if you think they are true or false? | Wave 63.5 (total n=1622) | | | | | | Wave 64 (total n=1841) | | | | | |
|---|---|---|---|---|---|---|---|---|---|---|---|---|
| | True | | False | | Don't know | | True | | False | | Don't know | |
| | % (95% CI) | n | % (95% CI) | n | % (95% CI) | n | % (95% CI) | n | % (95% CI) | n | % (95% CI) | n |
| You must wear a face covering in shops (unless you are exempt)* | 91.9 (90.5 to 93.2) | 1490 | 5.6 (4.5 to 6.7) | 91 | 2.5 (1.8 to 3.3) | 41 | 90.3 (88.9 to 91.6) | 1622 | 5.4 (4.4 to 6.5) | 100 | 4.3 (3.4 to 5.2) | 79 |
| You must wear a face covering on public transport (unless you are exempt)* | 91.1 (89.7 to 92.5) | 1477 | 6.2 (5.0 to 7.3) | 100 | 2.8 (2.0 to 3.6) | 45 | 91.7 (90.5 to 93.0) | 1689 | 4.8 (3.8 to 5.8) | 88 | 3.5 (2.6 to 4.3) | 64 |
| You must wear a face covering while moving around in restaurants, cafés and pubs (unless you are exempt)* | 64.5 (62.2 to 66.8) | 1046 | 28.2 (26.0 to 30.4) | 457 | 7.3 (6.1 to 8.6) | 119 | 71.2 (69.1 to 73.3) | 1311 | 19.5 (17.7 to 21.3) | 359 | 9.3 (8.0 to 10.6) | 171 |
| You must wear a face covering in all crowded and enclosed spaces where you come into contact with people you don't usually meet (unless you are exempt)† | 77.9 (75.8 to 79.9) | 1263 | 15.2 (13.4 to 16.9) | 246 | 7.0 (5.7 to 8.2) | 113 | 83.5 (81.8 to 85.2) | 1538 | 10.3 (8.9 to 11.7) | 190 | 6.1 (5.0 to 7.2) | 113 |
| All contacts of suspected Omicron cases must self-isolate, regardless of their vaccination status‡ | 80.1 (78.1 to 82) | 1299 | 9.1 (7.7 to 10.5) | 148 | 10.8 (9.3 to 12.3) | 175 | 76.9 (75.0 to 78.8) | 1416 | 12.7 (11.1 to 14.2) | 233 | 10.4 (9.0 to 11.8) | 192 |
| You should stay at home as much as you can* | 61.7 (59.3 to 64.1) | 1001 | 27.2 (25.0 to 29.4) | 441 | 11.1 (9.6 to 12.6) | 180 | 69.5 (67.4 to 71.6) | 1280 | 20.4 (18.6 to 22.3) | 376 | 10.0 (8.7 to 11.4) | 185 |
| You should work from home if possible* | 69.5 (67.3 to 71.8) | 1128 | 20.2 (18.3 to 22.2) | 328 | 10.2 (8.8 to 11.7) | 166 | 90.4 (89.0 to 91.7) | 1664 | 5.6 (4.6 to 6.7) | 104 | 4.0 (3.1 to 4.9) | 73 |
| You cannot meet other people indoors, unless you live with them, or they are part of your support bubble* | 38.1 (35.7 to 40.5) | 618 | 49.1 (46.6 to 51.5) | 796 | 12.8 (11.2 to 14.5) | 208 | 36.1 (33.9 to 38.3) | 665 | 49.8 (47.5 to 52.1) | 917 | 14.1 (12.5 to 15.7) | 259 |
| International arrivals must take a PCR test by the end of the second day after arrival and self-isolate until they receive a negative result* | 84.0 (82.2 to 85.8) | 1363 | 7.6 (6.3 to 8.9) | 123 | 8.4 (7.0 to 9.7) | 136 | 81.4 (79.6 to 83.2) | 1499 | 6.5 (5.3 to 7.6) | 119 | 12.1 (10.6 to 13.6) | 223 |
| You must wear a face covering at the cinema or theatre§ | – | – | – | – | – | – | 85.2 (83.5 to 86.8) | 1568 | 7.4 (6.2 to 8.6) | 137 | 7.4 (6.2 to 8.6) | 136 |

Bold answers are correct.
*Previously a rule used to prevent the spread of SARS-CoV-2 in England.
†Not previously a rule used to prevent the spread of SARS-CoV-2 in England.
‡Rule introduced to prevent the spread of the Omicron variant of concern.
§Previously a recommendation, but not a legal obligation, used to prevent the spread of SARS-CoV-2 in England.

**Table 4** Satisfaction with government response to Omicron

| | The government is putting the right measures in place to protect the UK public from the Omicron variant of coronavirus, % (n) | | I have enough information from the government and other public authorities on the symptoms of the Omicron variant of coronavirus, % (n) | | I have enough information from the government and other public authorities on how effective current vaccines are against the Omicron variant of coronavirus, % (n) | | I have enough information from the government and public authorities about what I can do to help prevent the spread of the Omicron variant of coronavirus, % (n) | |
| | Wave 63.5 (total n=1622) | Wave 64 (total n=1841) | Wave 63.5 (total n=1622) | Wave 64 (total n=1841) | Wave 63.5 (total n=1622) | Wave 64 (total n=1841) | Wave 63.5 (total n=1622) | Wave 64 (total n=1841) |
|---|---|---|---|---|---|---|---|---|
| Strongly agree | 12.5 (203) | 12.2 (224) | 12.5 (203) | 13.1 (242) | 12.0 (195) | 15.6 (287) | 18.6 (301) | 18.3 (336) |
| Agree | 34.3 (557) | 31.6 (581) | 33.7 (546) | 36.9 (680) | 36.1 (585) | 39.5 (728) | 49.3 (799) | 49.6 (913) |
| Neither agree nor disagree | 23.0 (373) | 22.8 (420) | 23.9 (388) | 21.6 (398) | 22.7 (369) | 21.0 (387) | 17.8 (288) | 18.2 (335) |
| Disagree | 17.3 (281) | 19.3 (356) | 21.9 (355) | 19.6 (360) | 19.1 (310) | 16.5 (303) | 9.2 (149) | 9.4 (173) |
| Disagree strongly | 10.9 (176) | 12.1 (222) | 6.5 (105) | 7.7 (142) | 8.2 (133) | 6.2 (115) | 4.2 (68) | 3.7 (69) |
| Don't know | 2.0 (32) | 2.1 (38) | 1.5 (25) | 1.0 (19) | 1.8 (30) | 1.1 (21) | 1.0 (17) | 0.8 (15) |
| Total strongly agree+agree, % (95% CI) | 47.8 (45.3 to 50.3) | 44.6 (42.4 to 46.9) | 46.9 (44.5 to 49.4) | 50.6 (48.3 to 52.9) | 49.0 (46.5 to 51.5) | 55.8 (53.5 to 58.1) | 68.5 (66.3 to 70.8) | 68.4 (66.3 to 70.5) |
| Total neither agree nor disagree+disagree+disagree strongly, % (95% CI) | 52.2 (49.7 to 54.7) | 55.4 (53.1 to 57.6) | 53.1 (50.6 to 55.5) | 49.4 (47.1 to 51.7) | 51.0 (48.5 to 53.5) | 44.2 (41.9 to 46.5) | 31.5 (29.2 to 33.7) | 31.6 (29.5 to 33.7) |

Engaging in highest risk social mixing and always wearing a face covering in hospitality venues were associated with worry about, and perceived risk of, COVID-19 (table 6). Always wearing a face covering in shops was independently associated with having heard more about Omicron. Associations between behaviour and Omicron-specific worry and perceived risk often did not reach the statistical significance level required after a Bonferroni correction but showed some relationship with behaviour. Always wearing a face covering was positively associated with having been vaccinated (see online supplemental table 2).

## DISCUSSION

These findings suggest that initial reporting of the emergence of Omicron had little impact on public perceptions. There were small increases in worry about, and perceived risk of, COVID-19 in the days after the emergence of Omicron was reported. While over one-third of participants reported being very or extremely worried about Omicron, and over half of respondents perceived a major or significant risk of Omicron to people in the UK, these figures were very close to the rates observed for concerns about 'coronavirus' in general.

Engagement with wearing a face covering and testing increased between 1 November and 16 December 2021. Approximately 80% of the sample reported 'always' wearing a face covering while in shops. This rate is similar to the percentage who reported 'frequently' or 'very frequently' wearing a face covering outside the home during the second lockdown in England (November 2020).[25] Rates of wearing a face covering increased even in hospitality settings, where rules were not changed, possibly reflecting the misunderstanding of the extent of official guidance that this study observed. A survey by the English Office for National Statistics also showed an increase in wearing a face covering in data collected 1 to 12 December 2021.[26] Increases in uptake of testing may reflect a higher prevalence of symptoms in the population during this period.[27] While there have been media reports of behaviour change in response to Omicron (eg, restaurant industry figures reporting a fall in eating out early on),[28] our results show that there were few changes in out-of-home activity up to 16 December 2021. This is in line with other polling carried out on 14 to 15 December 2021.[29] Despite Omicron being a key story in the media, it appears that early behavioural responses to it were largely restricted to changes that were required by legislation, rather than more spontaneous changes among the public.

Despite over one-third of people thinking that indoor mixing with other households was not allowed, there were no changes in patterns of social mixing. Our question on knowledge of the rules may be insensitive to degrees of certainty or may be demonstrating a social desirability effect. Social mixing may normally increase in the run-up to Christmas, so we cannot tell whether a flat statistic

**Table 5** Associations between out-of-home activities and amount heard about omicron, perceived worry, risk to self and risk to people in the UK

| Attribute | Level | Going out shopping (for groceries/pharmacy and other items) | | | | Attending the workplace | | | |
| | | Wave 63.5‡ | | Wave 64§ | | Wave 63.5¶ | | Wave 64** | |
| | | aIRR for going out shopping (95% CI) | P value | aIRR for going out shopping (95% CI) | P value | aIRR for attending the workplace (95% CI) | P value | aIRR for attending the workplace (95% CI) | P value |
|---|---|---|---|---|---|---|---|---|---|
| Amount heard about Omicron variant* | I have not seen or heard anything(1) to I have seen or heard a lot(4) | 1.05 (0.97 to 1.14) | 0.23 | 1.07 (1.00 to 1.16) | 0.05 | 1.02 (0.85 to 1.23) | 0.82 | 1.03 (0.87 to 1.21) | 0.77 |
| Worry about COVID-19* | Not at all worried (1) to extremely worried(5) | 0.92 (0.87 to 0.97) | 0.004 | 0.96 (0.91 to 1.01) | 0.08 | 1.06 (0.93 to 1.21) | 0.40 | 1.01 (0.89 to 1.14) | 0.93 |
| Worry about Omicron variant† | Not at all worried (1) to extremely worried (5) | 0.94 (0.86 to 1.02) | 0.15 | 0.93 (0.86 to 1.01) | 0.09 | 0.93 (0.77 to 1.11) | 0.40 | 0.95 (0.79 to 1.16) | 0.63 |
| Amount heard about Omicron variant* | I have not seen or heard anything (1) to I have seen or heard a lot (4) | 1.05 (0.97 to 1.14) | 0.26 | 1.08 (1.00 to 1.16) | 0.04 | 1.04 (0.87 to 1.25) | 0.65 | 1.02 (0.86 to 1.21) | 0.79 |
| Perceived risk of COVID-19 to self* | No risk at all (1) to major risk(5) | 0.95 (0.90 to 1.00) | 0.05 | 0.99 (0.94 to 1.04) | 0.65 | 0.97 (0.86 to 1.08) | 0.54 | 1.02 (0.92 to 1.14) | 0.68 |
| Perceived risk of Omicron variant to self† | No risk at all(1) to major risk (5) | 1.00 (0.91 to 1.09) | 0.95 | 1.02 (0.94 to 1.11) | 0.59 | 1.08 (0.92 to 1.28) | 0.35 | 1.04 (0.85 to 1.28) | 0.70 |
| Amount heard about Omicron variant* | I have not seen or heard anything (1) to I have seen or heard a lot (4) | 1.04 (0.96 to 1.13) | 0.30 | 1.07 (0.99 to 1.15) | 0.08 | 1.04 (0.86 to 1.24) | 0.70 | 1.02 (0.86 to 1.21) | 0.81 |
| Perceived risk of COVID-19 to people in UK* | No risk at all (1) to major risk (5) | 0.96 (0.91 to 1.02) | 0.24 | 0.98 (0.93 to 1.03) | 0.45 | 1.01 (0.89 to 1.16) | 0.83 | 0.98 (0.87 to 1.10) | 0.74 |
| Perceived risk of Omicron variant to people in UK† | No risk at all (1) to major risk (5) | 0.92 (0.84 to 1.00) | 0.05 | 1.10 (1.02 to 1.19) | 0.02 | 1.11 (0.94 to 1.31) | 0.24 | 1.01 (0.85 to 1.21) | 0.88 |

*Adjusting for all other sociodemographic characteristics; amount heard about Omicron, and worry about COVID-19/perceived risk of COVID-19 to self/perceived risk of COVID-19 to people in the UK.
†Adjusting for all other sociodemographic characteristics; amount heard about Omicron, and worry about COVID-19/perceived risk of COVID-19 to self/perceived risk of COVID-19 to people in the UK; and Omicron-specific worry/perceived risk to self/perceived risk to people in the UK.
‡A total of 1622 people were eligible for inclusion in analyses investigating going out shopping analyses. There were different amounts of missing data depending on variables included in the models, so n ranged between 1440 and 1491 for different models.
§A total of 1841 people were eligible for inclusion in analyses investigating going out shopping analyses. There were different amounts of missing data depending on variables included in the models, so n ranged between 1671 and 1713 for different models.
¶A total of 374 people were eligible for inclusion in analyses investigating non-essential workplace attendance (sample limited to people who reported they could work entirely from home). Due to missing data, n included in analyses ranged between 349 and 354.
**A total of 410 people were eligible for inclusion in analyses investigating non-essential workplace attendance (sample limited to people who reported they could work entirely from home). Due to missing data, n included in analyses ranged between 379 and 389.
aIRR, adjusted incidence rate ratios.

**Table 6** Associations between highest risk social mixing and wearing a face covering and amount heard about Omicron, perceived worry, risk to self and risk to people in the UK

| Attribute | Level | Highest risk social mixing | | | | Always wearing a face covering in shops | | | | Always wearing a face covering in hospitality venues | | | |
| --- | --- | --- | --- | --- | --- | --- | --- | --- | --- | --- | --- | --- | --- |
| | | Wave 63.5‡ | | Wave 64§ | | Wave 63.5¶ | | Wave 64** | | Wave 63.5†† | | Wave 64‡‡ | |
| | | aOR for engaging in highest risk social mixing (95% CI) | P value | aOR for engaging in highest risk social mixing (95% CI) | P value | aOR for wearing a face covering in shops (95% CI) | P value | aOR for wearing a face covering in shops (95% CI) | P value | aOR for wearing a face covering in hospitality venues (95% CI) | P value | aOR for wearing a face covering in hospitality venues (95% CI) | P value |
| Amount heard about Omicron variant* | I have not seen or heard anything (1) to I have seen or heard a lot (4) | 1.03 (0.88 to 1.20) | 0.72 | 1.08 (0.94 to 1.24) | 0.30 | **1.46 (1.19 to 1.79)** | <0.001 | 1.32 (1.09 to 1.59) | 0.004 | 1.12 (0.89 to 1.40) | 0.34 | 1.25 (1.02 to 1.53) | 0.03 |
| Worry about COVID-19* | Not at all worried (1) to extremely worried (5) | **0.79 (0.71 to 0.88)** | <0.001 | **0.73 (0.66 to 0.80)** | <0.001 | **1.43 (1.24 to 1.65)** | <0.001 | 1.44 (1.26 to 1.64) | <0.001 | **1.55 (1.31 to 1.84)** | <0.001 | **1.34 (1.17 to 1.55)** | <0.001 |
| Worry about Omicron variant† | Not at all worried (1) to extremely worried (5) | **0.76 (0.65 to 0.89)** | 0.001 | 0.93 (0.79 to 1.09) | 0.35 | 1.24 (1.00 to 1.53) | 0.05 | 1.35 (1.09 to 1.68) | 0.006 | 1.18 (0.95 to 1.47) | 0.13 | 1.26 (1.02 to 1.56) | 0.03 |
| Amount heard about Omicron variant* | I have not seen or heard anything (1) to I have seen or heard a lot (4) | 0.99 (0.85 to 1.15) | 0.88 | 1.03 (0.89 to 1.18) | 0.69 | **1.50 (1.22 to 1.84)** | <0.001 | 1.33 (1.10 to 1.61) | 0.003 | 1.17 (0.93 to 1.46) | 0.17 | 1.32 (1.08 to 1.62) | 0.007 |
| Perceived risk of COVID-19 to self* | No risk at all (1) to major risk (5) | **0.85 (0.76 to 0.94)** | 0.001 | **0.78 (0.71 to 0.86)** | <0.001 | **1.39 (1.21 to 1.60)** | <0.001 | **1.25 (1.10 to 1.41)** | <0.001 | **1.35 (1.16 to 1.57)** | <0.001 | **1.24 (1.09 to 1.41)** | 0.001 |
| Perceived risk of Omicron variant to self† | No risk at all (1) to major risk (5) | 0.85 (0.72 to 0.99) | 0.04 | 0.91 (0.78 to 1.05) | 0.20 | 1.20 (0.97 to 1.48) | 0.09 | 1.23 (1.00 to 1.50) | 0.05 | 1.26 (0.99 to 1.60) | 0.06 | 1.03 (0.83 to 1.27) | 0.81 |

Continued

**Table 6** Continued

| Attribute | Level | Highest risk social mixing | | | | Always wearing a face covering in shops | | | | Always wearing a face covering in hospitality venues | | | |
|---|---|---|---|---|---|---|---|---|---|---|---|---|---|
| | | Wave 63.5‡ aOR for engaging in highest risk social mixing (95% CI) | P value | Wave 64§ aOR for engaging in highest risk social mixing (95% CI) | P value | Wave 63.5¶ aOR for wearing a face covering in shops (95% CI) | P value | Wave 64** aOR for wearing a face covering in shops (95% CI) | P value | Wave 63.5†† aOR for wearing a face covering in hospitality venues (95% CI) | P value | Wave 64‡‡ aOR for wearing a face covering in hospitality venues (95% CI) | P value |
| Amount heard about Omicron variant* | I have not seen or heard anything (1) to I have seen or heard a lot (4) | 1.00 (0.86 to 1.17) | 0.98 | 1.03 (0.90 to 1.19) | 0.65 | **1.53 (1.25 to 1.87)** | **<0.001** | 1.35 (1.12 to 1.63) | 0.002 | 1.16 (0.93 to 1.45) | 0.19 | 1.32 (1.08 to 1.63) | 0.007 |
| Perceived risk of COVID-19 to people in UK* | No risk at all (1) to major risk (5) | **0.82 (0.73 to 0.91)** | **<0.001** | **0.83 (0.76 to 0.92)** | **<0.001** | **1.28 (1.11 to 1.49)** | **0.001** | **1.41 (1.23 to 1.61)** | **<0.001** | **1.41 (1.20 to 1.67)** | **<0.001** | **1.28 (1.11 to 1.48)** | **0.001** |
| Perceived risk of Omicron variant to people in UK† | No risk at all (1) to major risk (5) | 0.93 (0.80 to 1.09) | 0.37 | 0.91 (0.78 to 1.05) | 0.20 | 1.33 (1.09 to 1.63) | 0.006 | 1.37 (1.12 to 1.68) | 0.002 | 1.42 (1.14 to 1.78) | 0.002 | 1.20 (0.98 to 1.46) | 0.08 |

Bolding denotes significant findings (p<0.002).

*Adjusting for all other sociodemographic characteristics; amount heard about Omicron, and worry about COVID-19/perceivedrisk of COVID-19 to self/perceived risk of COVID-19 to people in the UK.

†Adjusting for all other sociodemographic characteristics; amount heard about Omicron, and worry about COVID-19/perceivedrisk of COVID-19 to self/perceived risk of COVID-19 to people in the UK; and Omicron-specific worry/perceived risk to self/perceived risk to people in the UK.

‡A total of 1622 people were eligible for inclusion in highest risk social mixing analyses. There were different amounts of missing data depending on variables included in the models, so n ranged between 1439 and 1446 for different models.

§A total of 1841 people were eligible for inclusion in highest risk social mixing analyses. There were different amounts of missing data depending on variables included in the models, so n ranged between 1668 and 1689 for different models.

¶A total of 1404 people were eligible for inclusion in wearing a face covering in shops analyses (sample limited to people who reported having been out shopping in the last week). There were different amounts of missing data depending on variables included in the models, so n ranged between 1247 and 1266 for different models.

**A total of 1600 people were eligible for inclusion in wearing a face covering in shops analyses (sample limited to people who reported having been out shopping in the last week). There were different amounts of missing data depending on variables included in the models, so n ranged between 1454 and 1475 for different models.

††A total of 789 people were eligible for inclusion in wearing a face covering in hospitality venues analyses (sample limited to people who reported having been out to hospitality venues in the last week). There were different amounts of missing data depending on variables included in the models, so n ranged between 700 and 713 for different models.

‡‡A total of 894 people were eligible for inclusion in wearing a face covering in hospitality venues analyses (sample limited to people who reported having been out to hospitality venues in the last week). There were different amounts of missing data depending on variables included in the models, so n ranged between 817 and 829 for different models.

aOR, adjusted OR.

actually represents a reduction compared with the likely pattern for the time of year. Nonetheless, in contrast to the early stages of the COVID-19 pandemic, we have not yet observed a substantial 'spill-over' effect involving non-recommended behaviours following the emergence of the Omicron variant.

Previous research has suggested that a constant stream of changes to guidance over the course of the pandemic left many people confused and disengaged.[30 31] Understanding of the new rules in response to Omicron was mixed. In general, people greatly overestimated the stringency of the rules. This had the potential to be positive in terms of reducing transmission, but also to have had a negative impact in terms of well-being,[32] economic activity[28] and social tension.[33] Additional rules were introduced on 13 December 2021 (England's 'plan B', working from home where possible, face coverings becoming compulsory in most public indoor venues apart from hospitality, introduction of vaccine passports in some settings).[11] Recognition of the rule regarding working from home increased in data collected 13 to 16 December 2021, but there was no evidence for a corresponding change in behaviour. This is likely because we measured behaviour in the previous week, before the rule was introduced. Furthermore, there was no legal underpinning to this rule in England, unlike during the third UK lockdown.[34]

We investigated associations between engaging in protective behaviours that had and had not been legislated for, and worry and perceived risk. Engaging in highest risk social mixing and always wearing a face covering in hospitality venues and while shopping were associated independently with worry about, and perceived risk of, COVID-19 in general. There were no associations for out-of-home activity (shopping and non-essential workplace attendance). Out-of-home activities may be perceived as being necessary (eg, shopping for provisions or attending the workplace at your employer's request). Results suggest that those behaviours that are perceived as being within one's control, such as wearing a face covering and engaging in risky social mixing, may be more affected by psychological factors.[35] Similar patterns of results and strengths of associations were seen for associations between behaviours and perceived risk to oneself and others in the UK. This is a slight difference to some previous research, which showed stronger associations between behaviour and perceived risk to others.[36 37] Of behaviours investigated, only wearing a face covering while shopping was a legislated behaviour. Wearing a face covering was also initially associated with having heard more about Omicron (wave 63.5). Data are cross-sectional and we cannot tell the direction of causation. It may be the case that people who wear face coverings are more likely to pay attention to news about COVID-19.

To the best of our knowledge, this is the first study investigating the influence of the Omicron variant on public worry, perceived risk and behaviour. This rapid response was facilitated by having regular data collection measuring public behaviour and attitudes. Limitations of the study include the use of self-reported data. We have previously noted that self-reported face covering wearing is likely to overestimate observed rates, although self-reports of 'always' wearing a face covering in a particular location appear more robust.[6]

Participants in our study were slightly more likely to be female, white and highly educated than the general population.[23 24] Whether the behaviour and attitudes of people who sign up to take part in surveys is representative of the behaviour and attitudes of the general population is unknown. Official statistics on uptake of the COVID-19 vaccine report percentages of the population aged 12 years and over.[38] Our sample comprised people aged 16 years and over and so are not directly comparable. Participants were asked to report on their behaviour in the last week. For wave 63, 63.5 and 64 data, this overlapped the period before and after rules (in response to the Omicron variant and England's 'plan B') came into force. We did not investigate factors associated with all potential out-of-home activities, nor uptake of testing, as this would have been too many outcomes. We focused our analyses on activities where the chance of coming into close contact with people from other households was greatest, and where legislation had recently changed. We investigated wearing a face covering only in people who reported having been out shopping or to hospitality venues in the past week. Workplace attendance was investigated only in those who reported being able to fully work from home. This limited our sample size and our ability to detect small effects. Data are cross-sectional and we are unable to determine direction of associations. One complicating factor for our analyses was the national discussion around 'partygate,' a news story that broke in November 2021 and was highly publicised in the following weeks, reporting on multiple occasions when government employees (including the Prime Minister) had attended gatherings that breached COVID-19 regulations.[39 40] This occurred at around the same time as the emergence of Omicron. A debate has developed over what, if any, effects the reporting about these social events had on public adherence.[41] We do not know if perceptions or behaviours might have been different, had reporting of these events not occurred at this time.

The Omicron variant emerged almost 2 years after the start of the COVID-19 outbreak. Despite substantial uncertainty about the impact of the resulting wave of infections, our data indicate that the emergence of the Omicron variant only slightly influenced worry about and perceived risk of COVID-19, suggesting a degree of habituation among the public to new announcements about the pandemic. Despite this, wearing a face covering, the main legislated change in response to Omicron, and uptake of testing increased between 1 November and 16 December 2021. These results suggest that specific behaviour changes continued to occur in response to changes in rules. Amount heard about Omicron was associated with always wearing a face covering, suggesting

that communications emphasising protective behaviours may also increase engagement for behaviours that are required by law.

**Author affiliations**
[1]Institute of Psychiatry, Psychology & Neuroscience, King's College London, London, UK
[2]NIHR Health Protection Research Unit in Emergency Preparedness and Response, King's College London, London, UK
[3]Institute of Health Informatics, University College London, London, UK
[4]Behavioural Science and Insights Unit, UK Health Security Agency, Salisbury, UK
[5]King's Centre for Military Health Research and Academic Department of Military Mental Health, King's College, London, UK
[6]Centre for Behaviour Change, University College London, London, UK

**Contributors** All authors conceptualised the study and contributed to survey materials. LES completed analyses with guidance from HP and GJR. LES and GJR wrote the first draft of the manuscript. HP, RA, NTF and SM contributed to subsequent drafts of the manuscript. LES, HP, RA, NTF, SM and GJR approved the final manuscript. GJR is guarantor. The corresponding author attests that all listed authors meet authorship criteria and that no others meeting the criteria have been omitted.

**Funding** This work was funded by the National Institute for Health Research (NIHR) Health Services and Delivery Research programme (NIHR project reference number (11/46/21)). Surveys were commissioned and funded by Department of Health and Social Care (DHSC), with the authors providing advice on the question design and selection. LS, RA and GJR are supported by the National Institute for Health Research Health Protection Research Unit (NIHR HPRU) in Emergency Preparedness and Response, a partnership between the UK Health Security Agency, King's College London and the University of East Anglia. RA is also supported by the NIHR HPRU in Behavioural Science and Evaluation, a partnership between the UK Health Security Agency and the University of Bristol. HP has received funding from Public Health England and NHS England. NTF is part funded by a grant from the UK Ministry of Defence. The Department of Health and Social Care funded data collection (no grant number).

**Disclaimer** The views expressed are those of the authors and not necessarily those of the NIHR, UK Health Security Agency, the Department of Health and Social Care or the Ministry of Defence.

**Competing interests** All authors have completed the ICMJE uniform disclosure form at www.icmje.org/coi_disclosure.pdf and declare: all authors had financial support from NIHR for the submitted work; RA is an employee of the UK Health Security Agency; HWWP received additional salary support from Public Health England and NHS England; HWWP receives consultancy fees to his employer from Ipsos MORI and has a PhD student who works at and has fees paid by Astra Zeneca; no other financial relationships with any organisations that might have an interest in the submitted work in the previous three years; no other relationships or activities that could appear to have influenced the submitted work. NTF is a participant of an independent group advising NHS Digital on the release of patient data. At the time of writing GJR is acting as an expert witness in an unrelated case involving Bayer PLC, supported by LS. All authors were participants of the UK's Scientific Advisory Group for Emergencies or its subgroups.

**Patient and public involvement** Patients and/or the public were not involved in the design, or conduct, or reporting, or dissemination plans of this research.

**Patient consent for publication** Not applicable.

**Ethics approval** This study involves human participants but this work was conducted as a service evaluation of the Department of Health and Social Care's public communications campaign. Following advice from King's College London Research Ethics Committee, it was exempt from requiring ethical approval. exempted this studyParticipants of online research panels have consented to being contacted to take part in online surveys. Following industry standards, consent was implied by participants' completion of the survey.

**Provenance and peer review** Not commissioned; externally peer reviewed.

**Data availability statement** No data are available. No additional data are available from the authors.

**ORCID iDs**
Louise E Smith http://orcid.org/0000-0002-1277-2564
Nicola T Fear http://orcid.org/0000-0002-5792-2925

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
