## [Reviewer comments · BMJ Open]

ARTICLE DETAILS

TITLE (PROVISIONAL)	How has the emergence of the Omicron SARS-CoV-2 variant of concern influenced worry, perceived risk, and behaviour in the UK? A series of cross-sectional surveys
AUTHORS	Smith, Louise; Potts, Henry; Amlot, Richard; Fear, Nicola; Michie, Susan; Rubin, G James

VERSION 1 – REVIEW

REVIEWER	Su, Zhaohui UT Health San Antonio, Center on Smart and Connected Health Technologies, Mays Cancer Center, School of Nursing
REVIEW RETURNED	27-Feb-2022

GENERAL COMMENTS	Congratulations and well done. My recommendation is "accept".
---

REVIEWER	Saxinger, Lynora University of Alberta Faculty of Medicine and Dentistry, Department of Medicine, Division of Infectious Diseases
REVIEW RETURNED	14-Mar-2022

GENERAL COMMENTS	This is a report from a longitudinal COVID-10 research panel survey programme which has been ongoing, in which the opportunity was taken to add specific additional survey questions to assess public worry, risk perception, behaviors and understanding and response to Government policy changes around the emergence of the omicron VOC. I'm reviewing this from the point of view of a knowledge user with Infectious diseases content expertise and extensive engagement in evidence synthesis to support policy in the pandemic, with some non extensive experience in survey-based research. I am not a statistician. Overall: This is a valuable dataset, and I think further distillation of the research findings and some additional editing would improve the strength of the paper. Some suggestions to consider if the data available allow: Population surveyed description- 1) The authors acknowledge the main methodologic challenge of those who choose to complete online surveys compared to the general population As readers might not be familiar with other work published around the survey, additional comments around
--

	representativeness of the panel if assessed against the population including the participant characteristics (unemployment, deprivation) would be appreciated. Additional comment on the proportion who believe themselves to have been infected and their reported vaccination status compared with the general population may also illuminate whether the panel reflects a group more likely to engage with vaccination and protective behaviours overall. 3) The interplay of these features as seen in trends in table one of the supplement may merit some comment in the main paper, for example the aIRR for attending the workplace, affected by educational attainment (apparent trend, NS) is a possible comment on work flexibility and white-collar jobs, with similar trends for language etc 4) In addition it seems potentially notable that the belief that one had COVID did not impact discretionary risk behaviours as much as vaccination status appeared to (although this was NS). 5) In supplement table 2 there appears to be potentially significant geographic variation may merit some comment in the main body as well General comments-questions on findings: 1) Is there data to assist sorting out whether individuals who've been vaccinated infected or both have a greater or lesser degree of worry impacting their behaviour? 2) The endorsement of rules section on page 12 seems to suggest a degree of confusion and some carryover assumptions from prior rules- would it be possible to indicate for an international audience which of the rules have been operational in the UK in the past, or if they all have at some time or another? 3) Is there a way to ascertain or comment on whether those who did not agree the government was putting into place the "right measures" felt that they were too relaxed or too stringent? Was there regional variation in this and did it track with behaviours such as mask wearing? 4) Table 5 seems to indicate activities which may be seen as routing-necessary, such as getting provisions do not change by own perceived risk while Table 6 suggests increasing mask use and decreased social mixing occurred against the evolving risk background. Table 6 also suggests that risk to self and risk to 'people in the UK' overall track quite similarly which I believe is a bit different than some other data which suggests altruistic motivation may be lower and this may be worthy of further comment. General comments: language and editing 1) "Beliefs about worry" is a phrase in the abstract and elsewhere and I think degree, or level of worry may be intended. 2) Additional attention to writing and editing is suggested. As an example, line 19 page 6 is a long, complex sentence and might be rephrased significantly, to something like: ' the survey encompassed behaviour over the previous week, thus wave 63 data included a short pre-omicron period. The added survey (wave 63.5) was issued after the emergence of omicron but encompassed a period before and after new rules became active.
--	--

	The wave 64 survey encompassed reported behaviour in the week before the new “Plan B” rules were instituted, with a small amount of representation of time under Plan B rules.' 3) Page 22 line 32- for international readers explaining 'partygate' and 'number 10' would be useful. 4) Page 9 line 41 - in this paragraph, the sentences are starting with numbers and this is overall hard to read. As the numbers refer to a table, consider something like: Thirty nine to 42.7% reported they were “very or extremely” worried”, and a higher proportion (56.7 to 61.4% perceived a major or significant risk to people in the UK than to themselves (44.9-46.4%). Figure 1 2 and 3 are useful overall but I'd consider altering the format to better be able to see trends as the small incremental percentages are not well shown on a long horizontal graph. Could you change the axes or show a delta from baseline instead?
--	--

VERSION 1 – AUTHOR RESPONSE

Reviewer: 1

Dr. Zhaohui Su, UT Health San Antonio

Comments to the Author:

Congratulations and well done. My recommendation is "accept".

We thank the reviewer for their comments.

Reviewer: 2

Dr. Lynora Saxinger, University of Alberta Faculty of Medicine and Dentistry Comments to the Author:

This is a report from a longitudinal COVID-10 research panel survey programme which has been ongoing, in which the opportunity was taken to add specific additional survey questions to assess public worry, risk perception, behaviors and understanding and response to Government policy changes around the emergence of the omicron VOC.

I'm reviewing this from the point of view of a knowledge user with Infectious diseases content expertise and extensive engagement in evidence synthesis to support policy in the pandemic, with some non extensive experience in survey-based research. I am not a statistician.

Overall:

This is a valuable dataset, and I think further distillation of the research findings and some additional editing would improve the strength of the paper.

We thank the reviewer for their comments.

Some suggestions to consider if the data available allow:

Population surveyed description-

1) The authors acknowledge the main methodologic challenge of those who choose to complete online surveys compared to the general population. As readers might not be familiar with other work published around the survey, additional comments around representativeness of the panel if

assessed against the population including the participant characteristics (unemployment, deprivation) would be appreciated. Additional comment on the proportion who believe themselves to have been infected and their reported vaccination status compared with the general population may also illuminate whether the panel reflects a group more likely to engage with vaccination and protective behaviours overall.

We have added additional comparisons between socio-demographic characteristics of the sample and the general population to the “respondent characteristics” section in the methods. For employment status, we found seasonally adjusted rates of employment in those aged 16 to 64 years (<https://www.ons.gov.uk/employmentandlabourmarket/peopleinwork/employmentandemployeetypes/timeseries/lf24/lms>), but as our sample includes participants aged 16 and over (no upper limit) and does not adjust for seasonality, we were unable to draw a direct comparison.

The true number of people who have been infected with SARS-CoV-2 is difficult to estimate. Previously, the ONS had used presence of antibodies as a marker of previous infection. This had its own problems with waning antibodies with increasing time from infection. However, with the introduction of widespread vaccination, rates of presence of antibodies is no longer a valid marker of previous infection. The ONS also estimate prevalence of cases during the pandemic. However, these estimates do not report cumulative cases. To the best of our knowledge, there is no accurate estimate of number of people who have been infected with SARS-CoV-2 since the start of the pandemic to compare our belief of having had COVID-19 measure to.

We have added vaccination status to respondent characteristics reporting. Population level statistics indicate that on 1 November 2021 (first day of data collection in study period analysed in this manuscript), 86.7%, 79.2%, and 14.5% of the population aged 12 years and over had received their first, second, and third/booster vaccine doses respectively. On 16 December 2021 (last day of data collection in study period analysed), 89.2%, 81.4%, and 45.6% of the population aged 12 years and over had received their first, second, and third/booster vaccine doses respectively. However, our sample comprises of people aged 16 years and over so rates are not directly comparable. On 1 November 2021, 22.0% of 12- to 15-year-olds had received one vaccine, and 0.3% had received two vaccines. On 16 December 2021, vaccine uptake had increased to 39.2% (one dose) and 0.7% (two doses) in this age group. We are unable to calculate how this affects percentage uptake in the general population aged 16 years and over. We have added a sentence to this effect in the limitations.

3) The interplay of these features as seen in trends in table one of the supplement may merit some comment in the main paper, for example the aIRR for attending the workplace, affected by educational attainment (apparent trend, NS) is a possible comment on work flexibility and white-collar jobs, with similar trends for language etc

Due to the large number of analyses run, we have chosen to take a cautious approach to our results, so as not to overstate our conclusions. As such, we have applied a Bonferroni correction to adjust for possible Type 1 errors. We have also based our interpretation on factors that are consistently associated with outcomes across time points. Therefore, we have not commented on results that do not reach our Bonferroni cut-off at one time point, such as the association between education and attending the workplace ($p=0.22$ in Wave 63.5, $p=0.004$ in Wave 64; Bonferroni cut off, $p<0.002$).

4) In addition it seems potentially notable that the belief that one had COVID did not impact discretionary risk behaviours as much as vaccination status appeared to (although this was NS).

While we agree with the reviewer, we have decided not to incorporate this point into the manuscript for the reasons outlined above.

5) In supplement table 2 there appears to be potentially significant geographic variation may merit some comment in the main body as well

We have not commented on slight potential geographic variation as there were no significant associations with behaviour, with the slight exception of always wearing a face covering in shops in Wave 64 data. However, as there was no overall geographic variation ($p=0.05$), and this pattern was only found in Wave 64 data, we have not highlighted this result in the manuscript.

General comments-questions on findings:

1) Is there data to assist sorting out whether individuals who've been vaccinated infected or both have a greater or lesser degree of worry impacting their behaviour?

Questions asking about vaccination and infection status do not ask when people received their vaccination, or they think they had COVID-19 and therefore may not be a good proxy for perceived immunity. While the data do allow us to compute a composite measure combining vaccination and infection status, and to investigate its association with worry, we have not done so for this Omicron-focused manuscript. An analysis exploring perceived immunity, based on a different dataset, is available elsewhere (<https://www.ncbi.nlm.nih.gov/pmc/articles/PMC7641362/>)

2) The endorsement of rules section on page 12 seems to suggest a degree of confusion and some carryover assumptions from prior rules- would it be possible to indicate for an international audience which of the rules have been operational in the UK in the past, or if they all have at some time or another?

We agree with the reviewer that there was confusion in understanding of the rules brought in to prevent the spread of the Omicron variant. We have added footnotes to the table to describe whether rules were new, or had been previously in place to prevent the spread of COVID-19 in England.

3) Is there a way to ascertain or comment on whether those who did not agree the government was putting into place the "right measures" felt that they were too relaxed or too stringent? Was there regional variation in this and did it track with behaviours such as mask wearing?

Unfortunately, there is no follow-up measure to indicate whether people who did not agree the government were putting "the right" measures in place thought the measures were too relaxed or too stringent. It is beyond the scope of this paper – which focuses on the Omicron variant of concern – to investigate associations between trust in government, region and engagement with protective behaviours.

4) Table 5 seems to indicate activities which may be seen as routing-necessary, such as getting provisions do not change by own perceived risk while Table 6 suggests increasing mask use and decreased social mixing occurred against the evolving risk background. Table 6 also suggests that risk to self and risk to 'people in the UK' overall track quite similarly which I believe is a bit different than some other data which suggests altruistic motivation may be lower and this may be worthy of further comment.

We have added to the discussion that there was no association between worry and perceived risk of COVID-19 and behaviours that may have been perceived as necessary (e.g. outings for shopping or to attend the workplace). Behaviours over which people may perceive having greater control (e.g. wearing a face covering and engaging in highest risk social mixing) may be more likely to be influenced by psychological processes.

As suggested by the reviewer, we have also added that wearing a face covering and engaging in highest risk social mixing were associated with perceived risk to oneself and people in the UK. This is different to previous research suggesting that uptake of protective behaviours during the pandemic was motivated by perceived risk to others.

General comments: language and editing

1) "Beliefs about worry" is a phrase in the abstract and elsewhere and I think degree, or level of worry may be intended.

We have amended this as suggested.

2) Additional attention to writing and editing is suggested. As an example, line 19 page 6 is a long, complex sentence and might be rephrased significantly, to something like: ' the survey encompassed behaviour over the previous week, thus wave 63 data included a short pre-omicron period. The added survey (wave 63.5) was issued after the emergence of omicron but encompassed a period before and after new rules became active. The wave 64 survey encompassed reported behaviour in the week before the new "Plan B" rules were instituted, with a small amount of representation of time under Plan B rules.'

We have amended this section, and the rest of the manuscript, to make the writing easier to read.

3) Page 22 line 32- for international readers explaining 'partygate' and 'number 10' would be useful.

We have given an explanation of these terms and used supporting references to make this sentence clearer to readers.

4) Page 9 line 41 - in this paragraph, the sentences are starting with numbers and this is overall hard to read. As the numbers refer to a table, consider something like: Thirty nine to 42.7% reported they were "very or extremely" worried", and a higher proportion (56.7 to 61.4% perceived a major or significant risk to people in the UK than to themselves (44.9-46.4%).

We have now amended this paragraph as suggested to make it clearer for the reader.

Figure 1 2 and 3 are useful overall but I'd consider altering the format to better be able to see trends as the small incremental percentages are not well shown on a long horizontal graph. Could you change the axes or show a delta from baseline instead?

We appreciate the reviewer's suggestion, but have decided to leave the figure axes as they are so

that they are easily accessible to all readers. We hypothesised that showing delta change would be more confusing to readers without specialised statistical knowledge.